# OpenReview forum: "RE-Bench: Evaluating Frontier AI R&D Capabilities of Language Model Agents against Human Experts"
_ICML.cc/2025/Conference — ICML 2025 spotlightposter_

### Official Review · Reviewer_ojQx · 2025-03-10

**Overall Recommendation:** 4

**Summary:**

The authors provide 7 research engineering problems together with surrounding environments. They evaluate both humans and AI models (Claude Sonnet 3.5, o1) on these problems, over different amounts of time spent and solution attempts. The results show that over short time horizons, the AI models tend to perform better, whereas humans gain more later on. Overall, the benchmark measures performance of AI agents in real-world engineering tasks of small scope. The goal is to get a sense of when AI models will be able to automate real-world engineering, with implications for the development speed of frontier AI.

**Claims And Evidence:**

Yes, the claims are supported by clear and convincing evidence.

**Essential References Not Discussed:**

I am not aware of essential references not discussed.

**Experimental Designs Or Analyses:**

I did not check the experimental designs or analyses in detail, but they seemed reasonable from reading the main paper.

**Methods And Evaluation Criteria:**

Yes, they make sense. Nevertheless, some notes on major limitations:

A. The main limitation: The engineering problems designed for this benchmark are relatively small-scale, such that humans can make substantial progress in the time frame of one day to one week. As the authors note, it is unclear to what extent the findings generalize to realistic engineering problems in frontier-model development, where the authors expect that AI currently performs much worse compared to humans than in the evaluations in this work.

B. One problem I see is that the benchmark can only test full automation of engineering tasks; thus, if the AI agent can only do 90% of the work to succeed at a task but completely fails at the remaining 10%, it would score zero in this benchmark, even though a human engineer having access to such a model could potentially significantly speed up their work. Thus, one should be cautious to get reassurance that AI R&D will *not* speed up based on limits in benchmark performance in such a benchmark. Future work could thus try to look into human-AI collaboration on engineering tasks (which would, however, probably be harder to evaluate and replicate).

C. Another potential limitation is that the two scaffolds used in this work seem relatively arbitrary, and the results show that different scaffolds can yield significantly different results. Thus, we could imagine that a third type of scaffold, which doesn't yet exist, could change the results substantially.

D. To get a sense of recent improvements, it would be interesting to know how well newer models perform (Gemini 2.0+Jules, Claude 3.7+Claude Code, o3-mini, DeepSeek-R1, ...).

**Other Comments Or Suggestions:**

Overall assessment: I currently rate this paper as "accept" as is, i.e., I think even without addressing the limitations I mention above, this is a good paper. I assume that this work must have been an immense effort and I appreciate that it gives us an indication of where current frontier AI stands in terms of ability to automate realistic engineering skills. The questions and concerns I mention might go significantly beyond the scope of this work.

Depending on the extent to which the limitations are addressed in the rebuttal, and depending on problems surfaced by other reviewers that I didn't catch (and that I consider significant), it is possible that I will increase or decrease the evaluation.

Typos:

a) p1, right side, 25: "programming, computer use, question answering" -- an "and" seem missing

b) p.8, left side, 38: "parallely-developed"

c) p.8, right side, 430: "the ability of increasingly capable to autonomously..." -- increasingly capable frontier models?

**Other Strengths And Weaknesses:**

Strength: The paper is very well written.

Weaknesses:

i. Figure 2 does not clarify immediately how the allocation of a total time budget to samples and a time horizon is done (though it seems like later you clarify that you use the optimal allocation)

ii. The main paper might benefit from more details on some or all of the 7 tasks, to get more of a feel for what the AI agents are evaluated for.

**Questions For Authors:**

1. The paper only evaluates fairly recent models. Do older models fail the tasks completely? If not, do we have any indication of a "trend"? I'd find it particularly interesting to know whether there is a trend toward AI agents beating humans over increasingly long time-horizon tasks, or for an increasing "engineering-cost".
2. How much money was spent on human and/or AI evaluations? How much tweaking is required to run your evaluations on new models? This could give other researchers an indication of whether they have the resources to use your evaluation environments for their own evaluations.
3. Do you have thoughts on whether strong RE-Bench performance would differentially lead to a stronger improvement in AI capabilities vs. AI safety?
4. How good are the human experts?
5. Claude 3.5 Sonnet curves upward toward the end in Figure 2. Any idea how to interpret this? (Though tbc., it probably wouldn't curve upward for a linearly-spaced x-axis)
6. Do you know what it is about the scaffolds that changes the results so much?
7. Could you provide an intuition for how to reconcile Figures 2 and 4 with each other, which look quite different from each other?

**Relation To Broader Scientific Literature:**

See the paper, Section 4 on related work.

**Theoretical Claims:**

The paper does not contain any theoretical claims.

---

> ### Author Rebuttal · Authors · 2025-04-01
>
> We thank the reviewer for the thorough review and insightful comments\! We are glad that the reviewer finds the work to be well-written, the evidence to be clear and convincing, and that the results provide insight into the capabilities of current frontier AI on realistic research engineering.
>
> We address the reviewer’s concerns in “Methods and Evaluation Criteria” below:
>
> 1) **We believe that RE-Bench represents a substantial improvement in the realism and complexity of AI-relevant research engineering evaluations compared to the existing literature (see Table 7 for a detailed comparison)**\! In designing this benchmark, we decided that 1-day long tasks would be a reasonable compromise between providing valuable new insights and the cost and complexity of running evaluations and getting high-quality human baseline data. However, we recognize that this is a limitation of our work and discuss how time-horizons may affect our results in Section 5\. We are excited for future work to build on RE-Bench and develop better ways of measuring longer-task performance\!
> 2) **We believe that RE-Bench can measure partial AI performance quite well**\! We carefully designed our tasks and score functions such that progress is intermediately measurable and it is possible to make easy and quick progress. For example, in the “Optimize a Kernel” task, a very easy improvement is to use built-in Pytorch functions to optimize the slow starting solution. In “Optimize LLM Foundry”, an easy optimization is removing unneeded steps like checkpointing. We thus think that AIs that can reach close to an expert solution for a task are also likely to find easier or partial solutions that make some measurable progress. We would love to see future work investigate how human-AI collaborations might score on tasks like these, and hope that RE-bench can facilitate this.
> 3) We aimed to be principled in our choice of scaffolds, with Modular representing a simple proven baseline, and AIDE representing a more complex open-source scaffold that achieved top scores on the closest existing benchmark (MLE-bench). We certainly agree that scaffolding is an unsolved problem in AI evaluations and we discuss some of these limitations in Section 5.3. We are excited for future work on better understanding their effects and **we hope that RE-Bench can help serve as a valuable testbed for other researchers\!**
> 4) While we agree that these results would be interesting, we note that the newer models were released less than 4 months before the ICML submission deadline, which falls under “concurrent work” according to the ICML reviewer guidelines. **Therefore, we think that these evaluations would be out-of-scope for our work**. However, we certainly are excited for future work that evaluates these, and newer, models on RE-Bench\!
>
> Here’s our responses to the reviewer's questions. Note, due to the character limit, we’ve had to be brief in our response.
>
> 1) Based on experimentation and qualitative judgements, we expect models substantially older than Claude 3.5 Sonnet (Old), which is the oldest model evaluated in our work, to perform very poorly in RE-Bench.
> 2) The average token cost for agent runs is \\$123, while we paid human experts \\$1855 on average; GPU costs vary but H100s generally cost around \\$2/hr (and tasks use 0-6 GPUs). However, we want to note that these environments could be run on less expensive GPUs (like A100s), but the collected run data would not be comparable. Close to no tweaking of the environments, and often none to scaffolds as well, are needed to run newer models.
> 3) RE-bench has been designed to measure capabilities research primarily.
> 4) We believe our professional network experts (scoring 0.98 on average) are very strong, with over 5 years of experience in an area highly relevant to the task they are baselining or recent experience at frontier ML research organizations. More information can be found in Section 2.1 or Appendix A\!
> 5) We suspect this result is due to noise.
> 6) Qualitatively, Claude 3.5 Sonnet (new), for example, seems to be better at tool-use (invoking the right tools correctly, and interpreting their results) than o1-preview, which affects how well models work with different scaffolds.
> 7) Figure 2 plots **time-budget** on the x-axis, i.e. increasing amounts of the total time available to agents, and each point is the average performance when using our best-observed way of allocating that time for that AI agent, which often involves doing BoK over many short runs. Whereas in Figure 4, we are plotting wall-clock time on the x-axis for a single run. Scores increase more slowly in this setting, because models often get stuck with bad solutions or incorrect assumptions over time. Frequent resetting seems to increase solution diversity and helps them continue to make progress.
>
> On the suggested improvements for the writing and the typos, we’ll address these in our updated draft of the paper\!

---

> > ### Comment · Reviewer_ojQx · 2025-04-01
> >
> > Thank you for the response! I have read this response, all other reviews, and your responses to the other reviews. I have not yet read any reviewer-responses to your responses since they don't yet exist.
> >
> > I think your answers are reasonable and I keep my score of 4. I think this is a good paper.

---

### Official Review · Reviewer_oWPp · 2025-03-14

**Overall Recommendation:** 3

**Summary:**

This paper contributes a new LLM (Agent) benchmark **RE-Bench**, consisting of 7 ML research engineering tasks for evaluating whether AI agents can autonomously perform AI R&D. A human study is also conducted, and results are analyzed.

## update after rebuttal

I thank the authors for their rebuttal. I maintain my score after reading all the other review comments.

**Claims And Evidence:**

* "Seven novel, hand-crafted evaluation environments covering realistic ML research tasks.". Yes this is substantiated.
* "Data from 71 attempts by 61 ML experts...results"; yes.
* "Qualitative analysis", was also substantiated by the evidence in the paper.

**Essential References Not Discussed:**

No essential references not discussed.

**Experimental Designs Or Analyses:**

* Tasks validated with pilot human tests to ensure feasibility.
* Multiple agent scaffolds and time horizons thoroughly compared.
* Agents repeatedly query the scoring function, letting them brute-force improvements quickly.
* Analyses are careful, e.g. acknowledging noise in QA fine-tuning.
* Overall design is robust and fair, though limited by having only 7 tasks.

**Methods And Evaluation Criteria:**

* Benchmark Construction: 7 custom tasks, each with a scoring function, a baseline, and a strong reference solution. Score is normalized between 0 (baseline) and 1 (reference).
* Human Comparison: 61 experts, each given 8 hours, same compute environment.
* AI Agents: Tested on the same tasks with up to 8 hours, plus best-of-k variants for shorter runs.
* Criteria: Normalized scores, time-based performance, best-of-k sampling.

**Other Comments Or Suggestions:**

* Clarify if any rule-breaking “cheating” solutions were excluded from final results.
* Discuss how specialized domain experts vs. generalized experts might affect human baselines.

**Other Strengths And Weaknesses:**

* Clarity: The paper is well written and clear to read.
* Originality: The proposed benchmark is novel and original.
* Significance: The benchmark focuses on solving a problem values by the community.

**Questions For Authors:**

* How were “cheating” or environment-breaking agent solutions handled in scoring?
* Did any participants use external LLMs while solving, and how did that affect results?

**Relation To Broader Scientific Literature:**

Positions RE-Bench relative to prior coding/ML agent benchmarks (MLE-bench, GAIA, etc.), highlighting its novelty in offering long-horizon tasks and direct human comparisons. Also references frontier AI policy frameworks. Good coverage of existing agent scaffolds.

**Theoretical Claims:**

Not applicable, as no new theoretical claims.

---

> ### Author Rebuttal · Authors · 2025-03-31
>
> We are pleased to hear that the reviewer found the paper is well-written and clear to read, RE-Bench is novel and original, and the contributions significant. We address the questions the reviewer had below and we’d be happy to provide any further clarifications.
>
> Q1: How were “cheating” or environment-breaking agent solutions handled in scoring?
>
> Any runs that we inspected that had evidence of cheating were excluded from the results entirely. For agent solutions, we carefully inspected the 2 best performing runs for all tasks, and we inspected many more of the top performing runs on “Restricted Architecture MLM” and “Optimize LLM Foundry”. These two tasks often had cheating because agents would often forget or not take into account the restrictions. When running these experiments and developing the environments, we have not seen cheating or environment-breaking agent solutions for any other tasks.
>
> For human runs, we manually inspected score logs and submissions for any signs of cheating or environment-breaking solutions.
>
> Q2: Did any participants use external LLMs while solving, and how did that affect results?
>
> Participants were allowed to use the internet, external LLMs and other tools in order to solve the task. We wanted to compare AI agents against humans with access to their preferred development environment and tools, as that would be more representative of how frontier research is conducted and provide the strongest baseline. To our knowledge, we expect most participants to have used LLM-based coding tools like Cursor, or web-based LLM interfaces. We would be excited to see future work explore how human performance on research tasks varies with access to different types of AI assistance.
>
> Additional discussion around both questions can be found in Appendix A for more details\!

---

### Official Review · Reviewer_ayPP · 2025-03-14

**Overall Recommendation:** 4

**Summary:**

In this work, the authors propose RE-Bench, which is designed to assess the capabilities of AI agents for AI research and development, especially in comparison with human experts. They define 7 ML engineering environments with scoring functions and evaluate human experts and AI agents on those under the same time budget. Under similar constraints with access to the scoring functions, they find that given a small amount of time, AI agents can have an edge over the human experts, which is significantly flipped given more time. They suggest that this can be a reasonable benchmark for evaluating and developing AI agents for research.

**Claims And Evidence:**

- I have some concern on the claim about "open-ended ML research engineering." Please refer to the Methods And Evaluation Criteria section for the details.

**Essential References Not Discussed:**

Excluding concurrent work, I believe this work fairly cited relevant work.

**Experimental Designs Or Analyses:**

- The experiments are rigorously done with well-designed setups. For instance, the use of Vivaria for providing VM environments with GPUs can be an important factor for reproducibility and reliability of the presented results.
- The authors provide a lot of technical details and decisions as well as empirical analyses. They can provide useful insights about the benchmark and especially the comparison of human experts and AI agents.

**Methods And Evaluation Criteria:**

- Using the time as the universal budget for both humans and AI agents is an interesting decision, because one of the major strengths of AI agents over humans is their computing powers.
- My primary concern about this work is the fact that the authors allow invoking the scoring function for evaluation without any limits ("The scoring function, which defines the goal of the environment, can be run by the agent at any time."). This could imply two things:
  - (a) Viewing this with the lens of sequential decision-making, unlimited access to the scoring function may mean access to the ground-truth value function (or some function that provides analogous information). Although the search space is still quite huge, this decision may give too much advantage to fast-iterators, which is usually not the case with challenging real-world AI R&D tasks.
  - (b) In many real-world AI *research* tasks, achieving some degree of generalizability is required, because research is usually not targeted for a specific narrow downstream task. For instance, researchers are not supposed to check the test performance until the research is finished. Based on this, I have some concern about claiming that the environments are for evaluating the capabilities of agents for open-ended ML "research" engineering.
- On the other hand, I think the suggested environments can be reasonable, non-trivial testbeds to assess engineering or development capabilities of AI agents.

**Other Comments Or Suggestions:**

N/A

**Other Strengths And Weaknesses:**

- The quality of the presentation of this work and the manuscript is high. It is easy to follow thanks to the organized information. Also, it provides much of the low-level details and analyses, which can be useful resources for readers and other researchers in the field.

**Questions For Authors:**

- The use of the time budget can be one of the choices for the "x-axis" for the evaluations and comparisons. Have you considered other choices, such as the number of score evaluations?

**Relation To Broader Scientific Literature:**

AI-based AI research agents are getting attention these days. The topic and the problem that this work tackles are relevant to the broader scientific literature.

**Theoretical Claims:**

There is not much of theoretical claims from this submission.

---

> ### Author Rebuttal · Authors · 2025-03-31
>
> We thank the reviewer for their time, attention and thoughtful feedback. We are glad that the reviewer finds the RE-Bench tasks to be nontrivial assessments of research engineering. We appreciate that the reviewer found the paper and presentation of the work to be high quality and the experimental results to be rigorous, well-designed and insightful.
>
> Q1: My primary concern about this work is the [... question omitted due to character limit ...] I have some concern about claiming that the environments are for evaluating the capabilities of agents for open-ended ML "research" engineering.
>
> Thank you for bringing this up\! We offer several points in response to this concern:
>
> * The 7 tasks vary significantly in how easily and quickly the ground-truth score can be checked, even though they all allow an unlimited number of invocations of the score command. In environments like ”Restricted Architecture MLM” and “Fix Embedding”, **good solutions require long training runs before they can be meaningfully scored (sometimes taking many hours)**, so in practice agents/humans only have one or a few attempts to train and score a final solution. While **in** **the “Scaling Law Experiment” environment, the agent can never see the ground-truth score** **of its solutions.** In both cases agents can check the score performance of smaller models or shorter training runs frequently, but then have to account for generalization.
> * For some tasks in real AI R\&D work, the **ground-truth score function is actually quite easy to have access to**, for example, the runtime is very accessible when trying to optimize code bottlenecks for large training runs. Our task “Optimize a Kernel”, where the goal of the task is to make a GPU kernel run as fast as possible, is a realistic example where it’s very natural to have quick access to the actual score. Therefore, we think it is important to include such environments in our benchmark.
> * The reason we give ground-truth feedback in so many of the tasks is to avoid reasonable misunderstandings about how the score will be measured, what kind of generalization is intended, or what kinds of solutions are allowed, which we have previously found can happen very easily even for human experts and can make task results uninformative.
> * Agents being fast-iterators is indeed a real phenomenon. We discuss in Section 3.4 that on average, agents run the score function between 25.3-36.8 times per hour, compared to 3.4 times for humans. This seems to help significantly on tasks like “Optimize a Kernel” where rapid iteration against ground truth is possible, but it is not clear that it provides any real advantages on tasks like “Fix Embedding” where proper score assessment requires long training runs and exercising judgment.
> * Lastly, we acknowledge that this is indeed a consideration, and we offer a discussion of this in our Section 5.3.
>
> Q2: The use of the time budget can be one of the choices for the "x-axis" for the evaluations and comparisons. Have you considered other choices, such as the number of score evaluations?
>
> Thank you for this question\! Here are some points we hope addresses the reviewer’s question:
>
> * **Indeed, we include a comparison between humans and agents with non-GPU cost (i.e. pay for humans and token costs for AIs) on the x-axis as Figure 8**.
> * Due to practical constraints, human baselines had to be conducted with a time-limit of 8 hours (see Section 2.1). We offer additional discussion about how this affects our results in Section 5.2.
> * We also wish to highlight that **RE-Bench is agnostic to what limit is used for agents or humans.** This flexibility means that RE-Bench can easily be used to explore future work on different comparative settings.
> * Additionally, agents and humans were asked to optimize performance relative to the time they used, so using time as the x-axis is more representative of their best performances. To properly investigate how score relates to number of score evaluations would require rerunning human and agent results with instructions to minimize unnecessary score invocations.
> * Lastly, while not an x-axis, we also include discussion about the number of score evaluations in Section 3.4.

---

> > ### Comment · Reviewer_ayPP · 2025-04-04
> >
> > I appreciate the authors for providing the detailed response to my review.
> >
> > While I have some remaining concerns regarding the claim of "open-ended ML research" engineering, the paper and author response discuss that aspect to a certain degree, and thus I am raising my score to 4.

---

### Decision · Program_Chairs · 2025-05-01

**Decision:**

Accept (spotlight poster)

**Comment:**

All reviewers are positive about this paper, and authors have engaged well with suggestions for improvement. The domain is important, namely empirical investigations of AI safety, and of the ability of LLMs to do creative intellectual work.